# Nanotechnology for Age-Related Macular Degeneration

**DOI:** 10.3390/pharmaceutics13122035

**Published:** 2021-11-29

**Authors:** Bo Yang, Ge Li, Jiaxin Liu, Xiangyu Li, Shixin Zhang, Fengying Sun, Wenhua Liu

**Affiliations:** 1Department of Anesthesiology, The Second Hospital of Jilin University, Changchun 130012, China; yangb20@mails.jlu.edu.cn; 2School of Life Sciences, Jilin University, Changchun 130012, China; lige18@mails.jlu.edu.cn (G.L.); liujiaxin@hrbmu.edu.cn (J.L.); lixiangyu19@mails.jlu.edu.cn (X.L.); shixin20@mails.jlu.edu.cn (S.Z.); sunfengying@jlu.edu.cn (F.S.)

**Keywords:** age-related macular degeneration (AMD), nano-drug delivery system, cyclodextrin, CRISPR/Cas9, adeno-associated virus (AAV), hyaluronic acid, non-viral vector

## Abstract

Age-related macular degeneration (AMD) is a degenerative eye disease that is the leading cause of irreversible vision loss in people 50 years and older. Today, the most common treatment for AMD involves repeated intravitreal injections of anti-vascular endothelial growth factor (VEGF) drugs. However, the existing expensive therapies not only cannot cure this disease, they also produce a variety of side effects. For example, the number of injections increases the cumulative risk of endophthalmitis and other complications. Today, a single intravitreal injection of gene therapy products can greatly reduce the burden of treatment and improve visual effects. In addition, the latest innovations in nanotherapy provide the best drug delivery alternative for the treatment of AMD. In this review, we discuss the development of nano-drug delivery systems and gene therapy strategies for AMD in recent years. In addition, we discuss some novel targeting strategies and the potential application of these delivery methods in the treatment of AMD. Finally, we also propose that the combination of CRISPR/Cas9 technology with a new non-viral delivery system may be promising as a therapeutic strategy for the treatment of AMD.

## 1. Introduction

Age-related macular degeneration (AMD), an eye disease with irreversible vision loss caused by retinal epithelial cell decay and retinal degeneration, is currently the third leading cause of blindness among the elderly [1]. The pathogenesis of AMD includes chronic inflammation, persistent oxidative stress, and accumulation of lipofuscin and nucleoli. In addition, some studies have suggested that estrogen exposure may reduce the risk of AMD due to its anti-oxidant and anti-inflammatory activities [2]. With the progression of the disease, AMD is divided into two major subgroups: wet AMD (wAMD) and dry AMD. Dry AMD usually progresses quite slowly, but there is no definite treatment method, which is mainly based on regular follow-up. However, if not treated in time, dry AMD may develop into wAMD with a blinding rate of over 80% [3,4]. The pathologic features of wAMD are choroidal neovascularization (CNV) invasion and Bruch’s membrane rupture, causing vascular leakage, bleeding and scarring, and ultimately visual loss [2].

AMD is a major posterior chamber disease associated with the elderly. Due to the special structure of the eye as a natural barrier to drug absorption, local non-invasive ocular drug delivery for anterior chamber diseases is not suitable for AMD. Depending on the route of administration, the adsorption barrier of the drug is also different. The barriers of the periocular administration route include tear barrier, corneal barrier, conjunctival barrier, elimination of aqueous humor circulation, and blood-retinal barrier (BRB). For diseases of the posterior segment of the eye, the permeability of topical drugs is low. In addition to BRB, the barrier for intravenous or oral administration also includes systemic absorption of drugs. However, only trace amounts of drugs can penetrate retinal pigment epithelial cells and capillary cells. At the same time, due to the systemic absorption of the drug, there are larger side effects [5,6]. Compared with other local administration methods, the drug is injected into the vicinity of the retinal tissue to produce the highest level of bioavailability in the posterior chamber group. Currently, vascular endothelial growth factor (VEGF) inhibitors are mostly used clinically to treat wAMD, including the intravitreal injection of ranibizumab, VEGF-Trap (Aflibercept injection) and the injection of the off-label drug bevacizumab (Table 1). Although these antibody drugs can significantly improve the visual prognosis of patients with wAMD, some patients will show resistance to anti-VEGF inhibitors. Due to the different types of anti-VEGF medications that have unique mechanisms of action, patients usually need to receive repeated treatment monthly. In addition, individual differences may result in different responses of patients to the diseases, which not only brings an economic burden to patients but may also cause the risk of serious complications due to the repeated intravitreal injection. The pain of injection and these unpredictable reactions greatly limit patient compliance. Therefore, it is necessary to find a new delivery system to decrease the number of administrations, reduce the risk of complications, and improve patient compliance. Furthermore, the unexpected response to treatment suggests that different mechanisms of action may play a synergistic role in these diseases, and may be why monotherapy is not sufficient to slow the progression of degenerative disease. The structure of the eye, the larger molecular weight, and anti-VEGF antibody drug efficacy being shorter, requires invasive injection through the vitreous cavity.

The nano-drug delivery system (NDDS) is a gelatinous system that can encapsulate lipophilic or hydrophilic drugs of different molecular weights. NDDSs mainly include liposomes, nanomicelles, nanoemulsions, and nanoparticles. Because of their small size, biodegradation, and surface modification, NDDS are widely used in drug delivery for ocular and other diseases. This review discusses the development of nano-drug delivery systems for AMD in recent years. We also propose some new gene therapy and targeting strategies for AMD.

## 2. The Major Disease Mechanisms of AMD (Age-Related Macular Degeneration)

Currently, the pathogenesis of AMD is not entirely clear, but it appears to involve dysregulation of angiogenesis, light damage, oxidative stress, inflammation, and immune dysfunction. However, the above mechanisms on the pathogenesis of AMD’s actual process are not isolated, they interact and affect each other throughout the progression of AMD.

### 2.1. Dysregulation of Angiogenesis

#### 2.1.1. Vascular Endothelial Growth Factor

The particularly important reason for the formation of choroidal neovascularization (CNV) is the unbalanced secretion of VEGF and anti-angiogenic factors [7]. Under normal physiological conditions, VEGF expression is relatively low in eye tissues. However, the expression level of VEGF will increase significantly under the conditions of ischemia, hypoxia, and inflammation, which will induce the formation of pathological blood vessels. Based on this theory, vascular endothelial growth factor is regarded as a potential therapeutic target for CNV therapy. Existing anti-VEGF drugs include VEGF aptamer drug, pegaptanib (Macugen^®^), the anti-VEGF antibody fragment drug, ranibizumab (Lucentis^®^), bevacizumab (Avastin^®^), and aflibercept (Eylea^®^). They have made certain progress in the clinical treatment of ocular neovascular diseases [8]. However, these anti-VEGF drugs usually require repeated intravenous injections because of the half-life limitation. A novel nano-delivery system loaded with therapeutic proteins such as ranibizumab can be continuously delivered in vivo. In addition, this new delivery system can avoid the huge risks and costs of frequent intravenous injections [9,10,11,12].

#### 2.1.2. Glutamine Synthetase

Glutamine synthetase (GS), encoded by the GLUL gene, is an ATP-dependent synthetase that has a main biochemical function to synthesize glutamine from other substances in the organism. Glutamine is a carbon and nitrogen donor produced by biomolecules and participates in REDOX homeostasis. Glutamine can be converted into glutamate and ammonia under the catalysis of glutaminase. Glutamate can also be converted to α-ketoglutarate, and it can also be converted to ornithine to produce polyamines and nitric oxide (NO), both of which are pro-angiogenic factors [13,14]. Studies have proved that endothelial cell (EC) metabolism is closely related to pathological angiogenesis [15,16,17,18]. Particularly, the unique glycolytic properties of EC are critical to cell proliferation, migration, and response to environmental changes [15]. Joyal et al. demonstrated that retinal ECs use glycolysis and fatty acid oxidation (FAO) to generate ATP [19]. They also found that free fatty acid receptor 1 (FFAR1) inhibits glucose uptake in the presence of free fatty acids, which would lead to a double shortage of lipid/glucose fuel and a decrease in α-ketoglutarate. At the same time, low α-ketoglutarate increased VEGF secretion and led to abnormal angiogenesis. According to reports, glutaminase 1 (GLS1) and fatty acid synthesis are also involved in ocular neovascularization in mice [20,21,22]. It has been found that depriving ECs of glutamine or inhibiting GS can inhibit angiogenesis, affect the migration of ECs, and reduce intraocular neovascularization (Figure 1) [22].

### 2.2. Light Damage and Oxidative Stress

Studies have shown that oxidative damage to retinal cells can accelerate the development of AMD (Figure 2) [6,23]. Light damage and oxidative stress are important components in the complex pathogenesis of AMD. For the two subtypes of AMD, dry AMD is characterized by the degradation and death of photoreceptors and retinal pigment epithelial cells; wet AMD is related to choroidal neovascularization (CNV) [24]. The pigmentation and rapid degeneration of the retinal pigment epithelium are common clinical features in the early stages of AMD. The PR and RPE in the retina have a high degree of metabolic activity, and they are extremely susceptible to the effects of oxidative stress and aging to cause rapid functional degradation (Figure 3) [25,26]. The retina is a high oxygen-consuming tissue rich in photosensitizers. Long-term light exposure of the eyes will cause lipofuscin to be photo-oxidized to promote DNA oxidation and cell apoptosis [27,28]. Therefore, as the antioxidant capacity of the macular disease site weakens, PR, retinal pigment epithelial cells, and chorionic capillaries will produce more damage [29,30]. In addition, the chronic increase of oxygen free radicals can also cause mitochondrial DNA (mtDNA) damage in RPE cells [31]. This process is also accompanied by changes in the integrity of the matrix the membrane, and the number of mitochondria [32]. Under stress conditions, the level of superoxide dismutase in mitochondria decreases, leading to increased superoxide anions, degeneration of RPE cells, thickening of Brunch’s membrane, and finally cell apoptosis. Under stress conditions, the level of superoxide dismutase in mitochondria decreases and the level of ROS increases. This leads to increased superoxide anion, degeneration of RPE cells, thickening of Brunch’s membrane, and apoptosis [33]. ROS will also react with nucleic acids, proteins and lipids, and induce the production of VEGF to form new fragile blood vessels [34].

### 2.3. Inflammation and Immune Dysfunction

As a part of the innate immune system, the complement system plays a vital role in maintaining the balance of the eye microenvironment [35]. According to reports, the complement immune system is related to the pathology of AMD. One of the outstanding features of early AMD is the accumulation of extracellular deposits between the retinal pigment epithelium and choline, called drusen genome-wide association studies (GWAS), suggesting that the complement cascade was involved in AMD disease progression. In the early stages of AMD, complement proteins will accumulate in drusen and Bruch’s membranes; simultaneously, the levels of C3a, C3Bb, and C5a in the patient’s plasma, aqueous humor, or vitreous fluid increase. In addition, new epitopes formed during oxidative stress in patients will bind to autoantibodies and activate the complement system. The accumulation of complement proteins in the body will increase white blood cells and inflammatory mediators, thereby further enhancing the local inflammatory state and driving the occurrence of AMD. Studies have confirmed that mutations in complement C3, complement factor I (CFI), complement factor H (CFH), and complement factor B (CFB) genes are closely related to the occurrence of AMD, and CFI is particularly significant [36,37]. In summary, these findings indicate that the dysregulation of the complement system is closely related to the pathogenesis of AMD. In addition, Johnson et al. believe that RPE dysfunction can cause inflammation and produce drusen, which activates the supplementary cascade to generate an immune response related to RPE [38]. Anderson et al. also found that the accumulation of drusen can cause a local chronic inflammatory response, which intensifies the stimulation of the primary disease [39]. The C3 inhibitor (APL) developed by Apellis Pharmaceuticals, Inc., and the C5 inhibitor (Zimura) developed by the Original Ophtothtech Corporation (NCT 02686658) prove that the complement system has potential application prospects as a drug target for the treatment of AMD.

## 3. Drug Delivery Systems for AMD

### 3.1. Liposomes

Liposomes are types of lipid vesicles with a diameter in the range of 0.1–10 microns, have strong encapsulation power, high biocompatibility, and can control the release of drugs. Therefore, they are considered suitable drug delivery vehicles (Figure 4A). The physicochemical properties (size, surface charge, and chemical properties) of liposomes can be changed by combining different lipid formulations. According to the research of Elsaid et al., the cholesterol–PEG liposomes containing rapamycin with small particle size and uniform distribution effectively solve the local drug delivery obstacles caused by poor water solubility [40]. Because liposomes can provide a variety of chemical sites for surface modification, control drug release according to the number and composition of lipid bilayers, and provide the potential to respond to drug release, etc., it is widely used in eye disease research. For example, Behroozi et al. reported a redox-sensitive smart liposome loaded with N-acetylcysteine. It was found that the expression of antioxidant genes in retinal pigment epithelial (hESC-RPE) cells increased. This new delivery system opened up a new direction for targeted therapy of retinal degeneration [41]. Joseph et al. reported that DPPC–DPPG liposomes loaded with ranibizumab released the drug more continuously than other formulations [42]. In addition, the bevacizumab loaded with multivesicular liposomes (Bev-MVLs), designed by MU et al., can continuously release bevacizumab and increase the intravenous retention time of the drug [43]. 

Currently, the standard treatment for AMD is administered by intravitreal injection. It is noteworthy that liposome-based delivery methods can avoid the high cost and risk of complications caused by repeated intravitreal injection. Karumanchi et al. prepared a liposome containing bevacizumab to decrease the cost and frequency of injections [44]. Platania et al. designed a small unilamellar lipid vesicle containing TGF-β1 supplemented with Annexin V and Ca2+. The evaluation proved that this new formula could skip invasive intraocular injections and reach the back of the eye [45]. Recently, gene therapy has been proposed as an extremely advantageous strategy for ocular diseases. Compared with other treatment methods, gene therapy is expected to express therapeutic proteins continuously, so as to broadly solve non-hereditary retinal diseases, such as AMD. Chen et al. proved that RGD-PEGylated liposomes loaded with VEGF-siRNA can effectively reduce the level of VEGF in ARPE-19 cells and have therapeutic potential for ophthalmic gene therapy [46]. Takashima et al. prepared pDNA/PEI composite liposomes, which showed high nucleic acid encapsulation efficiency and cell absorption capacity in ARPE-19 cells, which significantly improved the efficacy of posterior gene therapy [47]. It is worth noting that liposomes injected intravenously into the eye will not distribute smoothly and remain in the extensive lymphatic vascular network [48]. Moreover, liposomes still have some limitations, including low encapsulation efficiency, poor stability and high cost. Therefore, the optimization of charge, size, and lipid formulation will be key factors to be considered when designing a drug delivery system for eye-targeting at a specific location [49]. For example, positively charged or micron-sized liposomes are more suitable for later stage diseases such as AMD [50].

### 3.2. Nanomicelles

Nanomicelles are ordered structures formed by the self-assembly of amphiphilic compounds in water, with polar groups facing outwards and non-polar groups facing inwards. When the concentration is greater than the critical micelle concentration, self-assembly occurs (Figure 4B). In ocular administration, nanomicelles have the advantage of increased epithelial permeability and almost no irritation due to their nanoscale size.

Ravid et al. studied polymer nanomicelles containing Dex with polyethylene glycol-poly (ε-caprolactone) diblock copolymer as carrier materials [51]. Compared with the suspension liquid phase of Dex, nanomicelles increased the permeability of Dex to conjunctiva and sclera of rabbits by a factor of 2 and a factor of 2.5, respectively. These results suggest that this nanomicelle preparation may provide sufficient Dex and other hydrophobic drugs to the posterior ocular tissues after topical administration. Ma et al. developed a cidofovir micelle preparation with lipid derivatives that could maintain an effective concentration for at least nine weeks after a single intravitreal injection, providing conditions for the continuous release of drugs for chronic retinal diseases contained in this preparation [52]. Alshamrani et al. prepared a curcumin (CUR-NMF) aqueous nanodrop formulation for post-ocular administration to treat AMD. The results showed that the CUR-MNF preparation had good oxidative stress protection and anti-VEGF effects in D407 cells [53]. A transparent tacrolimus nanomicelle formulation (TAC-NMF) developed by Gote et al. can reduce the levels of pro-inflammatory cytokines and ROS to prevent AMD from being induced by inflammation [54]. However, nanomicelles also have disadvantages, including poor stability and fast drug release. Some degradation products of polymeric micelles are toxic or inflammatory to the sensitive ocular tissues [55]. Therefore, the clinical applicability and local administration method of the delivery system still need to be studied in depth.

### 3.3. Nanoemulsions

Nanoemulsions are transparent or translucent homogeneous dispersions of oil-in-water (O/W) or water-in-oil (W/O) with diameters of 1~100 nm. Nanoemulsions have good tissue permeability and can effectively promote the absorption of drugs into the tissue. Therefore, smaller doses of drugs can be used to achieve faster treatment results, reduce side effects, and improve patient compliance (Figure 4C). Hagigit et al. found that nanoemulsions loaded with antisense oligonucleotides targeting VEGF-R2 can significantly inhibit the formation of neovascularization in the rat cornea and the vitreous of ROP mice, of vital importance in the clinical treatment of ocular neovascular diseases [56]. However, the disadvantage of nanoemulsions is that they are not suitable for long-term sustained release. Situ gel is a polymer solution that gelates after entering the body due to the phase change properties of the polymer. Therefore, by adding the nanoemulsion to the in-situ gel carrier, the release of the nanoemulsion can be prolonged and the therapeutic effect can be improved. To this end, Patel et al. designed an in situ nanoemulsion poloxamer gel to treat various ocular inflammations, which is different from traditional ophthalmic suspensions. The results showed that compared with other similar preparations on the market, this preparation did not irritate rabbit eyes and successfully improved the bioavailability of ophthalmic drugs [57]. Ge et al. prepared a lutein nanoemulsion in situ gel (P-NE-GEL) modified by osmogen. They found that P-NE-GEL can protect retinal pigment epithelial cells from photooxidative damage and reduce the rate of apoptosis and ROS levels [58]. The lutein-loaded nanoemulsion (NE) composed of isopropyl myristate, triacetin, Tween 80, and ethanol, prepared by Lim et al., increased the solubility and permeability of the drug. It is a promising delivery system for the treatment of AMD [59]. In addition, Laradji et al. prepared an osmotic composite and red oxide reactive nanogel containing hyaluronic acid to ensure triggered release and further delivery of the active agent to the back of the eye. This nanogel can deliver drugs to the back of the eye and has the potential to treat AMD [60]. Du et al. constructed a novel Chinese medicine microemulsion original glue. The results show that the in-situ gel can effectively deliver the microemulsion to the back of the eyes of AMD model rats through the cornea-living-retina route. In short, this new ophthalmic preparation provides a favorable research basis for the treatment of AMD [61]. Certainly, in-situ gels also have some limitations. For example, it may undergo sol-gel transition when injected into the eye due to physical and chemical stimuli [62,63]. This can cause blurred vision and sticky eyelids after the infusion of the drug [64].

### 3.4. Nanoparticles

In recent years, the prospects of nanoparticle systems are particularly noteworthy in drug delivery systems developed for AMD. Nanoparticles can be divided into natural polymer nanoparticles, synthetic polymer nanoparticles, metal oxide nanoparticles, silicon dioxide nanoparticles, and heavy metal nanoparticles according to the type of carrier materials. Nanoparticles can be used to encapsulate chemical drugs, protein drugs, nucleic acid drugs, etc. Polymer nanoparticles are composed of natural macromolecular albumin, gelatin, alginate, and chitosan or synthetic macromolecular polylactic acid (PLA), polymethyl methacrylate (PMMA), polylactic-glycolic acid (PLGA), and polyvinyl alcohol (PVA), etc. These nanoparticles have different particle sizes, shapes, and stability properties. Among them, aliphatic polyester polymers, such as PLGA and PLA, have been widely used as new delivery carriers because of their good stability, biocompatibility, and biodegradability (Figure 4D). PLGA polymetic nanoparticles can reduce the dosage and frequency of administration by improving and controlling the release of drugs, thereby reducing eye irritation. Bolla et al. designed a PLGA–PEG–biotin nanoparticle to enhance the absorption of lutein by retinal cells to treat AMD, which can better deliver drugs to the back of the eye [65]. Narvekar et al. prepared a PLGA nanoparticle to continuously release axitinib to reduce the frequency of intravenous injection. The results show that the anti-angiogenesis ability of this axon-loaded PLGA nanoparticle is significant [66]. Liu et al. prepared a PEI/PLGA nanoparticle (eBev-DPPNs) co-loaded with bevacizumab and dexamethasone electrostatically for the combination therapy of angiogenic ocular disease [67]. Studies have shown that using PLGA-NPs as a delivery vehicle for bevacizumab has a higher packaging efficiency and can improve the treatment of AMD [68,69,70,71]. The Fe3O4/PEG–PLGA polymer nanoparticles prepared by Yan et al. and Mab/PEG-conjugated gold nanoparticles designed by Hoshikawa et al. are extremely advantageous for the treatment of age-related macular degeneration [11,12]. On the other hand, PLA has been proved to be effective in targeting specific eye tissues, which provides possibilities for the application of aliphatic polyester nanoparticles in intraocular diseases. Bourges et al. prepared PLA nanoparticles containing fluorescent dyes. Four months after vitreous administration, fluorescein could still be detected in rat retinal pigment epithelial cells [72]. The polylactic acid/polylactic acid-polyethylene oxide nanoparticles (PLA/PLA-PEO) developed by Kim et al. can penetrate the retina and localize to the RPE. This slow controlled release system removes the half-life limitation of the water-soluble integrin antagonist peptide C16Y (C16Y-NP) [73]. The NP-[CPP] nanoparticle prepared by Wang et al. modified the PEG-PLA chain with cell penetrating peptide (CPP) is also a strategy for non-invasive treatment of CNV and enhancing drug accumulation [74]. However, polymer nanoparticles undergo transformation after long-term storage, leading to drug release [75]. In addition, it also has disadvantages such as high cost and the burst effect [76].

### 3.5. Cyclodextrin

Cyclodextrin (CD) is a type of cyclic polysaccharide compound with a hydrophilic outer shell and a closed hydrophobic inner cavity [77]. The hydrophobic inner cavity of CDs can encapsulate objects of different polarities, while the water-soluble shell can increase the solubility of drugs in polar media. Therefore, nanosystems based on cyclodextrins (CDs) can deliver hydrophobic drugs to the back of the eye for local treatment. Kam et al. applied 2-hydroxypropyl-β-cyclodextrin to an elderly mouse model, which significantly reduced Aβ by 65% and inflammation by 75% within three months. It is also applicable to CFH gene knockout mouse models [78]. Qian et al. prepared a delivery system composed of graphene quantum dots (GQD) encapsulation, and supramolecular β-cyclodextrin that co-loaded Bevacizumabu and Rani Bizumabu. The data shows that the β-cyclodextrin-based nanocarrier system can significantly increase the delivery rate and biocompatibility of AMD drugs. El-Darzi et al. showed that 2-Hydroxypropyl-beta-cyclodextrin (HPCD) is an effective delivery tool to reduce drusen deposition and inhibit the occurrence of AMD [79]. These studies show that cyclodextrin is an extremely promising AMD delivery system. These studies show that cyclodextrin is an extremely promising AMD delivery system. It is worth noting that high concentrations of cyclodextrin may cause greater toxicity in ophthalmic applications. Therefore, the design needs to focus on the formulation to reduce the effective concentration of cyclodextrin required. For example, incorporation of drug-cyclodextrin complexes into liposomes is a possible future option [80].

### 3.6. Dendrimers

Dendrimers are a macromolecular polymer with a size ranging from 10 to 100 nm. Drugs can be trapped in dendritic macromolecules by hydrogen bonding, hydrophobicity, and ion interactions, or they can be bound to dendritic macromolecules by covalent bonding. The terminal structure of dendrimers has amine, hydroxyl, and carboxyl functional groups and can be used for conjugated targeting ligands (Figure 4E). Because dendrimers have a unique terminal functional group structure (amine, hydroxyl, carboxyl, etc.), they can be used as conjugated targeting ligands with good physical and chemical properties and high biocompatibility. Marano et al. used lipophilic amino-acid dendrimers to deliver VEGF) oligonucleotides (ODN-1) to the eyes of rats. They found that this amino-acid dendrimer can enhance the delivery of ODN-1 gene and inhibit laser-induced CNV [81]. Yavuz et al. used PAMAM dendrites to deliver DEX conjugates and found that PAMAM can improve drug release time and permeability [82]. In addition, dendrimers can also be combined with lipid systems. Lai et al. designed a complex lipid system coated with PAMAM G3.0. Experiments have shown that the PAMAM G3.0-coated composite liposomes become more permeable and protect human retinal pigment epithelial cells and rat retina from photooxidative damage [83]. Therefore, the PAMAM G3.0 lipid system shows potential applications in various eye diseases Studies have shown that PAMAM dendrimers increase the permeability of drugs mainly by interacting with the cornea or loosening the connection of epithelial cells [84]. This process of changing the corneal barrier may increase the risk of vision loss.

### 3.7. Composite Drug Delivery System

Occasionally, a single drug delivery system may have problems, such as high burst rate, short release period, easy migration of particles, and high toxicity. Therefore, the composite delivery system has received extensive attention from scientists due to its simple design, strong slow-release capability, and low biological toxicity. For example, Hsu et al. developed a nanoparticle-hydrogel composite drug delivery system composed of polylactide-glycolide (PLGA) nanoparticles and chemically cross-linked hyaluronic acid to reduce the frequency of drug delivery [8]. The data showed that the carrier could slowly release the drug at a stable rate under physiological conditions, and it also could penetrate deeply into the retinal layer to improve the bioavailability of the drug. Xin et al. encapsulated a type of nanoparticle encapsulating ACD molecules in calcium alginate hydrogel. They proved that this eyedrop effectively reduced the release rate of ACD and inhibited angiogenesis in the AMD model [85]. Hirani et al. developed a composite drug system composed of PEG–PLGA nanoparticles (NPs) and PLGA–PEG–PLGA thermoreversible gel to load triamcinolone acetonide. The results show that they can help build new AMD treatment strategies [86]. Wang et al. found that combining oxide nanoparticles (GCCNPs) with a glycol chitosan shell and alginate gelatin-based hydrogel can produce a synergistic antioxidant effect and repair retinal pigment epithelial cells and photoreceptor cells faster. This combination therapy technique has the potential possibility to treat AMD [87]. Osswald et al. proved that the microsphere–hydrogel composite system could effectively control the release of the model drugs, such as anti-VEGF and ranibizumab, which may have significant advantages in the treatment of posterior ocular diseases [88,89,90].

## 4. Novel Treatment for AMD

Since anti-VEGF therapy was used in clinics, researchers have been looking for better treatment regimens, and gene therapy is expected to be a better alternative to long-term anti-VEGF treatment. Gene therapy is clinically used to treat various diseases, among which cancer and genetic diseases are its main application fields [91,92,93,94,95]. Compared with VEGF inhibitors and chemotherapies, gene therapy not only improves the method of lifelong drug therapy but may also directly cure the disease, with minimal side effects on normal tissues during the treatment process. Gene therapy is considered to be a therapeutic technique with good application prospects because of its advantages, such as strong pertinence, high targeting, remarkable effect, non-toxicity, and good tolerance. Gene therapy mainly includes two branches: gene replacement/enhancement therapy and gene editing therapy. Using vectors (such as adeno-associated virus AAV) to deliver small interfering RNA (siRNA) to specific sites is a powerful method in gene replacement/enhancement therapy. Next, siRNA can target and cleave complementary mRNA to induce post-transcriptional gene silencing. Gupta et al. have reported clinical trials of this method for the treatment of AMD [96].

Gene editing is another branch of gene therapy. Compared with other gene editing technologies, such as ZFN and TALEN, the CRISPR-Cas9 system has the advantages of high efficiency, high cost performance, and good knockout effect. The CRISPR/Cas system, derived from the adaptive immune system of bacteria and archaea, implements genome editing by inducing DNA double-strand breaks (DSBs) at target genomic sites. The type II CRISPR/Cas system is the simplest and consists of DNA endonuclease Cas9, CRISPR RNA (crRNAs), and trans-activated crRNAs (tracrRNAs). During application, crRNAs and tracrRNAs were fused to form a single chimeric guide RNA (gRNA). VEGF-A is crucial in the pathogenesis of AMD. Yiu et al. used a lentil vector loaded with CRISPR-Cas9 to infect ARPE-19 cells. The results showed that the levels of VEGF-A protein and angiogenesis were reduced [97]. Kim injected VEGFA-specific Cas9 nucleoproteins (RNPs) into AMD mice and found that Cas9 RNP effectively reduced the area of laser-induced choline vascularization (CNV) in the model [98]. Research by Huang et al. showed that CRISPR-Cas9 can be used to disrupt oxygen-induced retinopathy (OIR) and angiogenesis in CNV mice [99]. Wu et al. also conducted similar experiments in human retinal microvascular endothelial cells (HRECs) through a two-way AAV system. The results showed that rAAV5-CRISPR/Cas9 completely blocked the formation of human retinal capillaries [100]. The CRISPR/Cas9 system has been widely used to edit the genomes of humans, animals, plants, and microorganisms.

Actually, the main limitation of the CRISPR/Cas system is that it may cause unnecessary mutations in partially homologous DNA sites (called off-target), thereby reducing the therapeutic application of the technology [101]. In addition to restricting the expression of Cas nuclease with unique promoters in retinal cells, researchers have also developed several methods to reduce off-target editing. For example, Ran et al. relied on nickase and the nicks of the two DNA strands produced by two offset GRNAs to efficiently repair the DSB at a specific site [102]. Recently, Vakulskas et al. introduced a single point mutation (R691A) in SpCas9 to reduce off-target editing [103]. Finally, Merienne et al. used a second gRNA targeting Cas9 ATG to prevent its translation and expression. This self-activated KamiCas9 system greatly reduces off-target editing activities [103,104]. Another major concern in the clinical application of CRISPR is that the expression of CRISPR-Cas9 may cause severe cytotoxicity and trigger an immune response in the host. So far, there is no clear report that it will cause tissue damage. In addition, the effectiveness of CRISPR depends on the type of delivery method, but this is still a challenge in the clinical use of CRISPR [105]. Most importantly, clinical application A safe CRISPR genome editing system without any side effects is needed.

## 5. Drug Delivery Systems for Gene Therapy

The lack of a safe and effective delivery system for CRISPR/Cas9 is a major challenge for clinical use. The CRISPR/Cas9 system is often transmitted in vivo by viral vectors, including adenovirus vectors, adeno-associated virus (AAV) vectors, and lentiviral vectors (LVs). However, the possible cytotoxicity, immunogenicity, and long-term expression of viral vectors remain obstacles to clinical application. AAVs can easily insert the genes they carry into the DNA of the host, and the insertion location is close to the genes regulating cell growth, which may increase the risk of liver cancer. Recently, several non-viral delivery methods, such as lipid nanoparticles and Cas9 protein/gRNA ribonucleoprotein complexes, have been developed, but their stability, safety, and delivery efficiency are limited.

### 5.1. Adeno-Associated Virus (AAV )

Adeno-associated virus (AAV), as a small non-pathogenic dependent virus, has shown great potential in terms of safety and stable gene expression in the retina. Maclaren explored four AAV generation structures optimized by CFI sequences and compared them with the most common human CFI sequences. In addition, the expression of CFI in the retina of C57BL/6 J mice was also studied [106]. The results showed that CFI is mainly expressed in the retinal pigment epithelium and photoreceptors. In addition, the secreted protein in the vitreous humor has also been confirmed to have functional activity.

Ling et al. co-delivered a lentivirus of the Cas9 mRNA of Streptococcus and a lentivirus of the guide RNA encoding VEGFA. They found that this co-delivery could knock-out 44% of the VEGFA in retinal pigment epithelial cells and reduce the area of choroidal neovascularization by 63%. This proves that this transiently expressed engineered lentivirus may serve as a new therapeutic strategy for neovascular retinal diseases [107]. Chung et al. studied two Cas9 endonucleases (i.e., SpCas9 and SaCas9) and evaluated the delivery efficiency and the genome editing rate of AAV vectors. The results show that SpCas9 has a higher genome editing rate and CNV suppression ability. In addition, lower levels of VEGF proved its knock-out efficiency depends on genome editing rather than viral transduction [108]. However, the non-target transduction of AAV, especially the vector based on the natural serotype with non-specific tropism, brings challenges to the delivery of CRISPR/Cas9 [109]. The high-dose compensation solution may increase the risk of genome editing in non-target tissues and also increase the risk of immune response to AAV capsids.

### 5.2. Exosomes

Exosomes are nanovesicles composed of lipid membranes with a diameter of 30–150 nm. They play an important role in intercellular communication and have important guiding significance for the diagnosis and treatment of clinical diseases. Since the function of exosomes and molecular composition depend on their different cell sources, the effects of exosomes from different types of cells are different. RPE-derived exosomes induced by oxidative stress have been shown to be capable of delivering proteins and miRNAs as carriers. Kang et al. performed proteomic analysis of ARPE-19-derived exosomes, and the results showed that the protein in RPE cells of AMD patients may be transferred to the aqueous humor through exosomes [110]. Biasutto et al. also found that water-insoluble receptor proteins in retinal cells can also be transported into the vitreous cavity through ARPE-19-derived exosomes [111]. Hajrasouliha et al. found that after subconjunctival injection of RAC-derived exosomes in CNV model mice, the drug can be quickly transported from the conjunctiva to the choroid and retina. The results show that RAC-derived exosomes can target macrophages and vascular endothelial cells involved in the formation of CNV, and also inhibit laser-induced CNV and retinal vascular leakage. Therefore, RAC-derived exosomes provide a promising strategy for AMD [112]. Exosome, as a promising target drug carrier, has also been widely studied in recent years, but its encapsulation efficiency for large nucleic acids is low. 

In recent years, researchers have attempted to prepare membrane hybrid exosomes (HEs) by mixing liposomes and exosomes. Unlike ordinary exosomes, HEs can effectively encapsulate larger plasmids, including CRISPR/Cas9 expression vector, and thus have broad application prospects in vivo gene editing. In various types of intraocular neovascularization, CNV is derived from the endothelial cells of the choroidal capillary network. Choroidal capillaries are permeable vessels that have walls which are composed of monolayer endothelial cells and pericytes. In addition, CNV invasion and penetration of the Bruch membrane can cause vascular leakage and bleeding, leading to high vascular permeability. Based on the above two points, the size of HEs approximately 200 nm can penetrate the vessel wall and reach endothelial cells. At present, the most common method of clinical administration is intravitreal injection, which can easily cause infectious endophthalmitis and bacterial endophthalmitis. Periocular injection of exosomes may be a safer route of administration. At the same time, the local administration of extraocular polymers of nano-scale exosomes also requires more in-depth exploration. Packaging the CRISPR/Cas9 system into exosomes is more challenging than other delivery vehicles [113]. Although plasmid DNA can be loaded in exons by electrofusion, the efficiency is usually very low [114]. At the same time, another key to exosome engineering is to target disease-related cells to develop appropriate targeting strategies.

### 5.3. Lipid Nanoparticles

A lipid nanoparticle is a cell-like membrane particle with a diameter of approximately 100 nm. Its composition includes neutral lipids, cationic lipids, and cholesterol. Lipid nanoparticles are the most advanced non-viral gene delivery system clinically. Compared with other non-viral gene delivery vectors, it has lower cytotoxicity and immunogenicity. Finn et al. proved that the lipid nanoparticle system can provide CRISPR/Cas9 components and is a promising and effective genome editing platform [115]. Sung et al. found that using amino lipid DNA nanoparticles to deliver the CRISPR/Cas9 system can make cells less toxic. At the same time, nanoparticles can efficiently express cas9 and sgRNA in cells, which improves the efficiency of gene editing [116]. Recently, some studies have also proved that lipid nanoparticles, as an effective gene delivery system, to show promising results in the treatment of retinal diseases and other diseases of the back of the eye [117,118,119]. In addition, the encapsulation efficiency and penetration of lipid nanoparticles are high, which provides advantages for the delivery of genes to cells behind the eye. Lipid nanoparticles do not contain any viral components, helping to minimize safety and immunogenicity issues. But it also has obvious shortcomings. First, the nanoparticles must escape from the endosome. In addition, the Cas9:sgRNA complex must transfer to the nucleus after escaping, which is also a potential point of failure [120]. Therefore, the delivery efficiency of the CRISPR/Cas9 module is usually not high. Since the transformation efficiency will be affected by the target cell type, the nature of the cargo, etc., intensive screening must be done to determine the best lipid type and formula for constructing its system.

## 6. Novel Targeting Strategies

### 6.1. Aptamer

Aptamers are a type of DNA or RNA oligonucleotides with a short single-stranded structure, which play a physiological role in vivo by binding with target proteins with a specific three-dimensional spatial structure through high affinity. As an aptamer rich in guanosine, AS1411 can form G-quad structure anti-nuclease degradation and has a specific affinity with nucleolin protein (NCL), which can enter cells and interfere with normal cell replication and proliferation. Recent studies have shown that AS1411 is also a potential drug for the treatment of CNV [121,122]. Vivanco-Rojas et al. demonstrated that local administration of AS1411 can reduce corneal neovascularization in the model. Mechanism studies have shown that AS1411 significantly inhibits the follow-up response triggered by human limbal stroma (HLSC) VEGF stimulation. In conclusion, this new study further confirms that AS1411 is an anti-angiogenic therapy with a novel mechanism [122]. At the same time, NCL can be highly expressed on the surface of the ECS membrane, and AS1411 can specifically recognize and bind NCL on the membrane surface.

### 6.2. RGD Peptide

The dysfunction of RPE and the release of VEGF to the choroid are the main causes of ocular CNV and AMD. Excessive VEGF can induce the formation of choroidal neovascularization in patients with age-related macular degeneration.

According to reports, integrin receptor αVβ3 is highly expressed in ocular neovascular tissues of AMD patients. In addition, the vascular tissues of patients with diabetic retinopathy also highly express αVβ3 and αVβ5. Therefore, integrin receptor ligands can be used as targets of drug delivery and achieve targeted delivery. Peptides containing the RGD sequence (Arg-gly-Asp) can bind specifically to integrin receptors. Studies have shown that RGD peptide binds to αVβ3 receptor specifically, so it is commonly used to purify integrin receptors [123]. Singh et al. developed intravenously injected gene delivery nanoparticles targeting the retina to treat CNV and designed PLGA nanoparticles with double modification of transferrin and RGD peptide to deliver anti-VEGF intracellular plasmid to the lesions of CNV. Compared with unmodified nanoparticles, nanoparticles functionalized with transferrin and linear RGD peptides can better deliver drugs to the retinal region, thereby increasing the intracellular gene expression of RPE, retinal vascular endothelial cells, and the outer segments of photoreceptors. In addition, the smaller CNV region in the rat model can inhibit the laser-induced CNV progress in the rodent model [124].

### 6.3. Hyaluronic Acid

The outer membrane (ELM) and the inner limiting membrane (ILM) are the physical barriers of the retina with a multilayered structure. The ELM has a pore size of 3–3.6 nm due to the presence of banded attachments, and the ILM is composed of a three-dimensional network structure with a pore size of 10–25 nm. As the main glial activating cells, Muller cells play an important role in both physiological and pathological processes of the retina. Research has indicated that hyaluronic acid (HA) and albumin surface nanocarriers could penetrate the entire retinal layer and enter the submembrane space. Muller cell studies have demonstrated that both hyaluronic acid (HA) and albumin surface nanocarriers can penetrate the retinal structure into the subretinal space.

Sim et al. studied and evaluated a polysiRNA complex containing anti-VEGF and a surface-adsorbed polyethyleneimine (PEI) and HA. Due to the presence of the HA outer layer, the polysiRNA polymers had a smaller size (260.7 ± 43.27 nm) and electric charge (–4.98 ± 0.47 mV). Because HA can interact with the CD44 receptor of the Muller cells to penetrate the retinal barrier and enter RPE cells. Human retinal pigment epithelial cells (ARPE-19) can easily absorb the polysiRNA complex nanoparticles in vitro, which can directly inhibit the translation of VEGF mRNA. Therefore, intravitreal injection of polysiRNA composite nanoparticles effectively overcame the barrier of the vitreous body and retina to deliver the model drugs to the subretinal space [125].

## 7. Conclusions

In this review, we summarized the latest delivery methods for age-related macular degeneration drugs. Several DDS systems currently studied for drug delivery in the posterior segment of the eye, such as liposomes, dendrimers, and cyclodextrins, have good bioavailability and stability. At the same time, these delivery systems can also reduce the number of ocular administrations, reduce the frequency of administration, and thereby alleviate the suffering of patients. In addition, this article also described novel AMD gene therapy strategies, such as the CRISPR/Cas9 system, which are highly targeted, effective, non-toxic, well tolerated, and are considered to have good applications as prospective treatment technology. The lack of a safe and effective delivery system for the CRISPR/Cas9 system is a major challenge for clinical applications. Therefore, combining a safe and efficient drug delivery system with CRISPR/Cas9 technology will hopefully serve as a promising therapeutic strategy in the treatment of posterior ocular diseases in the future.

## Figures and Tables

**Figure 1 pharmaceutics-13-02035-f001:**
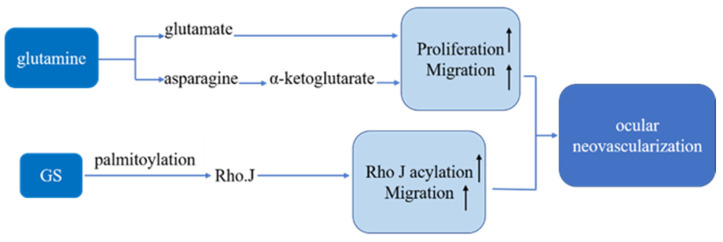
Model for the regulation of endothelial cell by glutamine and glutamine synthetase.

**Figure 2 pharmaceutics-13-02035-f002:**
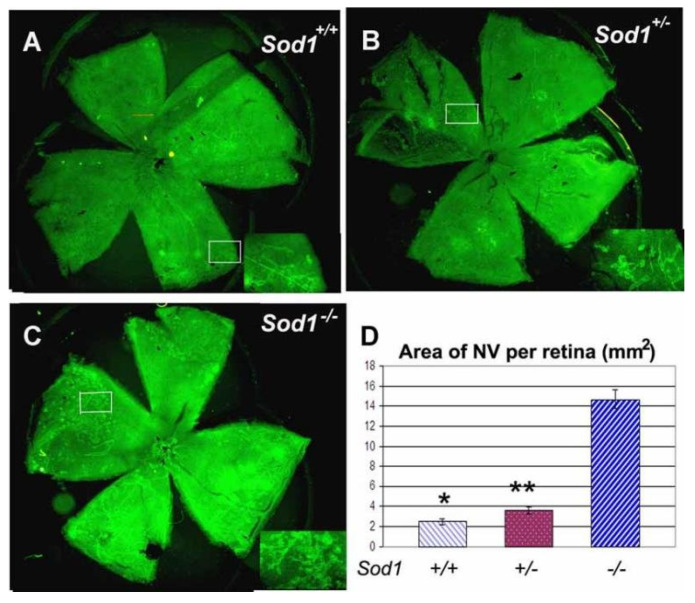
Distribution of choroidal neovascularization (NV) in Sod1−/− mice. Mice deficient in superoxide dismutase 1 (Sod1−/−) show excessive amounts of ischemia-induced retinal neovascularization compared to Sod1+/− or Sod1+/+ mice. Litters containing Sod1−/−, Sod1+/−, and Sod1+/+ pups were placed in 75% oxygen at postnatal day (P) 7, returned to room air at P12, and euthanized at P17. Retinal neovascularization on the surface of the retina was visualized by staining for PECAM-1 as described in Sod1+/+ (**A**) and Sod1+/− mice (**B**), but in comparison, Sod1−/− mice appeared to have substantially more neovascularization (**C**). Insets show a high magnification view of the retinal neovascularization present within the box in the whole mounts. Measurement of the area of neovascularization per retina by image analysis with the investigator masked with respect to genotype showed a marked increase in the mean (±SEM) area of neovascularization in Sod1−/− mice (n = 7) compared to Sod1+/+ (n = 10) and Sod1+/− (n = 8) mice (**D**). * *p* = 0.0005; ** *p* = 0.0005 by ANOVA with Dunnett’s correction for multiple comparisons. (Adapted with permission from [23], John Wiley and Sons and Copyright Clearance Center 2009).

**Figure 3 pharmaceutics-13-02035-f003:**
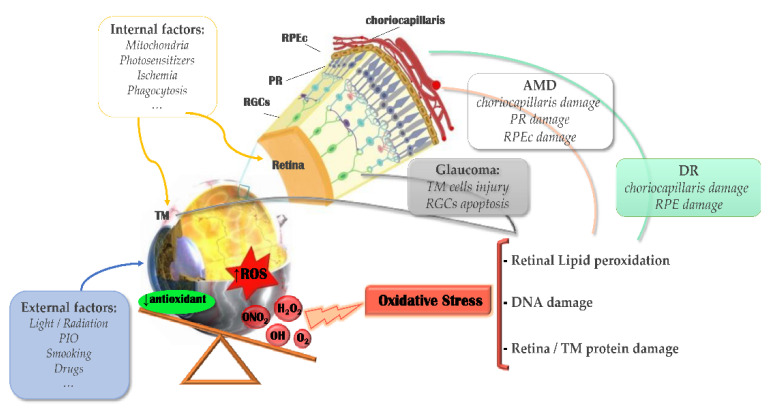
Scheme of oxidative stress role in posterior ocular diseases. IOP: intraocular pressure; ROS: reactive oxygen species; MT: trabecular meshwork; RGCs: retinal ganglion cells; PR: photoreceptors (cones + rods); RPE: retinal pigment epithelium; AMD: age-related macular degeneration; DR: diabetic retinopathy. (Adapted with permission from [25], Multidisciplinary Digital Publishing Institute 2021).

**Figure 4 pharmaceutics-13-02035-f004:**
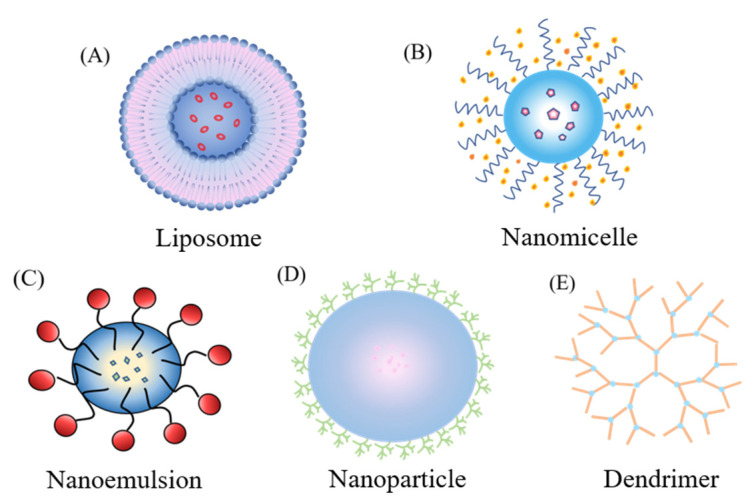
Schematic diagram of common drug delivery systems.

**Table 1 pharmaceutics-13-02035-t001:** Summary of current marketed drugs for the treatment of AMD.

Drug	Product	Time and Indication	Target	Company
Brolucizumab	Beovu^®^	7 October 2019 America, wAMD 17 February 2020 EU, wAMD	VEGFA	Alcon Laboratories; Novartis
Bevacizumab	Avastin^®^	26 February 2004 America, wAMD 26 February 2010 China, wAMD	VEGFA	Genentech; Chugai Pharmaceutical
Ranibizumab	Lucentis^®^	30 June 2006 America-Genentech, wAMD, RVO, DME, mCNV 22 January 2007 EU-Novartis, wAMD, DME 21 January 2009 Japan-Novartis, AMD, DME, CNV, RVO 31 December 2011 China-Novartis, wAMD, RVO, CNV	VEGFA	Genentech; Novartis; Roche
Aflibercept	Eylea^®^	18 November 2011 America-Regeneron Pharmaceuticals, wAMD 28 September 2012 Japan-Bayer, AMD, macular edema 21 November 2012 EU-Bayer, wAMD, DME, macular edema 2 February 2018, 30 November 2018 China-Bayer, DME, wAMD	VEGFA/B, PGF	Regeneron Pharmaceuticals; Bayer
Conbercept	Langmu^®^	27 November 2013 China wAMD 24 May 2017 China pmCNV 17 May 2019 China DME	VEGFA/B, PGF	KangHong Pharmaceutical Group; RemeGen
Pegaptanib Sodium	Macugen^®^/PrMacugen^®^	17 September 2004 America, wAMD 31 January 2006 EU, wAMD	VEGF	Valeant
Anecortave Acetate	Retaane^®^	16 December 2005 Australia, auxiliary ranibizumab treatment wAMD	VEGF	Alcon Laboratories
Verteporfin	Visudyne^®^	16 December 1999 Switzerland-Novartis 12 April 2000 America-Valeant 27 July 2000 EU-Novartis 16 October 2003 Japan-Novartis 15 May 2015 China-Novartis, PM, AMD	Photosensitizers	Novartis, Novelion, Therapeutics, Valeant
VistaMR	Vistaplus^®^	2014, AMD	MR	Eye Co

## Data Availability

Due to this article is a review, no data analysis is designed.

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
