# Peer review of "Nanotechnology for Age-Related Macular Degeneration"

_pharmaceutics, 2021, doi:10.3390/pharmaceutics13122035_

Round 1
Reviewer 1 Report
The manuscript Nanotechnology for age related macular degeneration is intended to describe the more recent advances in nanotechnology for treating this degenerative eye disease. I have to admit that gathering all the figures of the article in a single section is not the preferred choice for me, but it´s okay if the EIC and the authors accept that solution. In general, the whole manuscript, and specially the introduction is quite difficult to follow. It consists on several paragraphs no linked through any story. Please, rewrite it and provide the justification of this review paper.
In addition, several issues must be addressed carefully:
P1L35, Figure 1. Figure 1 is actually the graphical abstract of Reference 3 and means nothing without explanation. Please, include apropriate justification for this figure. Otherwise, remove it.
P1L44: Figure 2 is just a graphic representation of the eye. Please, discusss the barriers to drug adsoprtion in terms of their administraiton route (topical, periocular, parenteral... administrations)
Section 3.1. This is a review article. Please include references to recent investigations concerning the point addressed in section 3
Section 3.2 Discuss properly the effect of the enzyme SOD, cytoplasmatic and mythochondrial and the implications in nanotechnologically driven drug delivery systems
P6L146-153: Include references to these studies
P6L164-P7L194. This seems a bunch of references with no sense. Please, construct an history to connect that. Same occurs in P7L204-L219
In P8, section 4.4, the authors described the multiple origin of nanoparticles for drug delivery. In the same page, L261 they center the application to PLA and PLGA due to their good biocompatibility and biodegradability and then, they go further in affirming that PLA has been proved effective in targeting specific eye tissues (they should comment more about that). Nevertheless, in their bunch of references, they only focus their attention in PLGA nanoparticles. It´s quite confusing since they previously defended the PLA. Please, justify the choice of PLGA as DDS in your review.
P9L301: Please, cite these studies
P9, section 4.6 and section 4.7 suffer from the same scarcity as previous section. A review article does not consist on a series of unconnected references.
P11 section 6.2. No references of works related with exosomes are described
P11L443: Please, cite these studies
P12L455: Please, cite these studies
P12L470-L475: Please, cite references. Is that the work of Lee et al (L476)?
Other minor points:
P6L174: Annexin V
P7L230: In situ gel
VEGFA should go in capital letters
Reviewer 2 Report
The authors describe the developments in nano-drug delivery systems and gene therapy strategies for age-related macular degeneration (AMD).
The authors also describe some new targeting strategies and the potential applications of this delivery method in the treatment of AMD.
For the past 10 years or so, AMD's treatment has been mainly vitreous injection. There are also implants in the vitreous. But the reviewer thinks that we need to change this way. The reviewer thinks the nano-technology method is very useful for making breakthroughs.
The reviewer thinks this manuscript is well written. The reviewer doesn't think it's necessary to point out any major improvements.
Minor Points
Line 287.
- Cyclodextrin (CD) is a type of cyclic polysaccharide compound with a hydrophilic 287 outer shell and a closed hydrophobic inner cavity [42].
The authors cite it as 42 references. Their group has reported good results using cyclodextrin eye drops for clinical application in human diabetic macular edema.
Topical dexamethasone γ-cyclodextrin nanoparticle eye drops increase visual acuity and decrease macular thickness in diabetic macular oedema.
Ohira A, Hara K, Jóhannesson G, Tanito M, Ásgrímsdóttir GM, Lund SH, Loftsson T, Stefánsson E.Acta Ophthalmol. 2015 Nov;93(7):610-5. doi: 10.1111/aos.12803. Epub 2015 Jul 23.PMID: 26201996
- This is an old paper, but how is the nicotinic acetylcholine receptor blocker currently evaluated?
Campochiaro et al. 14 showed that administration of topical mecamylamine, a nonspecific nicotinic acetylcholine receptor blocker, has positive effects in patients with DME.
Campochiaro PA Shah SM Hafiz G . Topical mecamylamine for diabetic macular edema. Am J Ophthalmol. 2010;149:839–851, e831.
Reviewer 3 Report
In this manuscript, nanotechnological methods in treating age-related macular degeneration (AMD) has been reviewed. While the drug delivery to the retina is a challenging and important topic, the manuscript contains some concerns. Please, find detailed comments below.
- Page 1, line 35: The sentence "However, if not treated in time, it may develop into wAMD..." needs clarification whether it means that dry AMD can develop into wet AMD or something else. As commonly known, there is no treatment for dry AMD. In addition, overall lack of therapy would probably have already resulted in much bigger proportion of wet AMD cases than currently known (10-15%).
- Table 1. Brolucizumab is permitted also in EU.
- Page 5, lines 112-113: Please, clarify the metabolization of endothelial cells to control angiogenesis.
- Topics are handled rather superficially. For example, light damage is much more versatile than presented in the manuscript.
- All claims must be justified either by references or own data. There are numerous facts(?) thorough the text lacking of references.
- Critical discussion should include all technologies, including CRISPR/Cas9 and gene therapy. AMD is a very multifactorial disease for which the effectiveness and usefulness of these complex methods should be carefully considered.
- The manuscript would benefit from a careful language revision.
Round 2
Reviewer 1 Report
The authors have appropriately answered all the questions raised in the first review. After minor English corrections, the Manuscript could be accepted for publication
Author Response
Dear Reviewer:
Thank you for your letter and comments concerning our manuscript entitled “Nanotechnology for age-related macular degeneration” (Manuscript ID: pharmaceutics-1400485). Those comments are all valuable and very helpful for revising and improving our paper. We have studied comments carefully and have made correction which we hope meet with approval. The main corrections in the paper and the responds to your comments are as flowing:
The authors have appropriately answered all the questions raised in the first review. After minor English corrections, the Manuscript could be accepted for publication.
A: Thank you very much for your approval of this review. According to your suggestion, we have made an English revision on the official website recommended by MDPI and obtained an English editing certificate(Please see the attachment).
Finally, we sincerely thank you for your precious time and suggestions for us.
Kind regards,
Bo Yang

Reviewer 3 Report
The manuscript has improved during the revision. It is still not correct to say that cells are metabolized and many sentences are still lacking of specified references. On page 13 (line 500), it should probably be “Gene therapy” instead of just “Gene”. Moreover, it is advantageous to list benefits of different techniques but critical discussion is still missing.
